# Nanoparticles suppress fluid instabilities in the thermal drawing of ultralong nanowires

Injoo Hwang[1,2,5], Zeyi Guan [1,5], Chezheng Cao[3], Wenliang Tang[1], Chi On Chui[4] & Xiaochun Li [1,3✉]

Ultra-long metal nanowires and their facile fabrication have been long sought after as they promise to offer substantial improvements of performance in numerous applications. However, ultra-long metal ultrafine/nanowires are beyond the capability of current manufacturing techniques, which impose limitations on their size and aspect ratio. Here we show that the limitations imposed by fluid instabilities with thermally drawn nanowires can be alleviated by adding tungsten carbide nanoparticles to the metal core to arrive at wire lengths more than 30 cm with diameters as low as 170 nm. The nanoparticles support thermal drawing in two ways, by increasing the viscosity of the metal and lowering the interfacial energy between the boron silicate and zinc phase. This mechanism of suppressing fluid instability by nanoparticles not only enables a scalable production of ultralong metal nanowires, but also serves for widespread applications in other fluid-related fields.

[1] Department of Mechanical and Aerospace Engineering, University of California, Los Angeles, CA, USA. [2] Division of Mechanical Convergence Engineering, Silla University, Busan, Republic of Korea. [3] Department of Materials Science and Engineering, University of California, Los Angeles, CA, USA. [4] Department of Electrical and Computer Engineering, University of California, Los Angeles, CA, USA. [5]These authors contributed equally: Injoo Hwang, Zeyi Guan. ✉email: xcli@seas.ucla.edu

Ultra-long metal ultrafine/nanowires (10s to 100s of nanometers in diameter) with embedded functionalities and their facile fabrication (e.g., thermal drawing of micro preform) have been long sought after as they promise to offer substantially improved performance in numerous applications such as micro-structured photonic crystal fibers[1], optical micro-/nano-fibers[2], electronics in fibers[3], fiber-based metamaterials[4,5], fibers as a novel platform for sensing devices[6], studying chemical reactions[7], multi-material functional fibers[8,9], and more recently fibers as a platform for fabrication of nanowires and nanoparticles[10–13]. However, most crystalline metal nanowires with high aspect ratio are beyond the capability of current manufacturing techniques[11], due to the fluid instability induced by a low viscosity of molten metals and the large interfacial energy with the cladding, despite that reliable drawing of indefinitely long amorphous semiconductor and polymer nanowires has been achieved[14,15].

Metal microwire fabrication by thermal drawing in a glass cladding, known as the Taylor-wire process, has been in practice for decades. As this conventional technique combines with the stack-and-draw approach, the fabrication of fine metal wires becomes possible[16]. However, little success has been reported to fabricate long wires with a diameter of <1000 nm out of metals, especially those of higher melting temperatures, such as gold (Au), copper (Cu), zinc (Zn), and their alloys. Thermally drawn continuous Cu and Cu alloy microwires of 4 μm in diameter have been demonstrated[17,18]. Au microwires of 4 μm diameter were fabricated over a length of several centimeters, and this continuous length shrank to 20 μm as their diameter was reduced to 260 nm[19]. Fabrication of discontinuous copper–phosphor (CuP) wire with a diameter of 500 nm and a length of tens of micrometers has been reported using borosilicate glass cladding[20]. Fiber drawing via laser-based heat source pulling of short pieces of

platinum (Pt) microwires has been used to fabricate quartz-sealed Pt nanowires. The resultant fibers were tapered down to 10 nm in diameter yet with a length of only 5 mm[21]. Alternatively, a polyol process, which is the synthesis of metal-containing compounds in ethylene glycol, was used to fabricate Ag nanowires with a length up to 230 μm and a diameter of 60–90 nm[22,23]. Low-melting-temperature metals such as tin (Sn), lead (Pb), bismuth (Bi), indium (In), and their alloys are mostly thermally drawn in polymer cladding. The smallest diameter reported for metal wires that can be reliably drawn into indefinitely long arrays is about 4 μm and is achieved from $Sn_{0.95}Ag_{0.05}$ alloy with PES cladding[11]. Droplets, discontinuities, and structural deformation would be observed upon further size reduction. Nevertheless, thermally drawn functional fibers embedding indium wires with a diameter approaching 1 μm have been demonstrated[24]. While low-melting-temperature Pb–Sn alloys and Bi nanowires with a diameter down to 50 nm were reported, no experimental evidence was provided to support their continuity over the claimed drawn length[25].

The fundamental reason that long metal nanowire cannot be fabricated is fluid instability, described by the Tomotika model. Tomotika model expresses the dispersion relation of a long cylinder of viscous fluid jet surrounded by another indefinitely long viscous medium[26], where instability was introduced by perturbation-caused equilibrium surface deformation[27], also called varicose perturbation. It analytically expresses the surface perturbation of the fluid jet as an amplified wave under ideal conditions. This model also defines the term "associated instability growth time," which describes the time length when the liquid thread breaks into droplets. Thus it could essentially guide the thermal fiber drawing for the fundamental optimization of drawing parameters for indefinitely long microwires/nanowires. Optimization of experimental settings, such as drawing

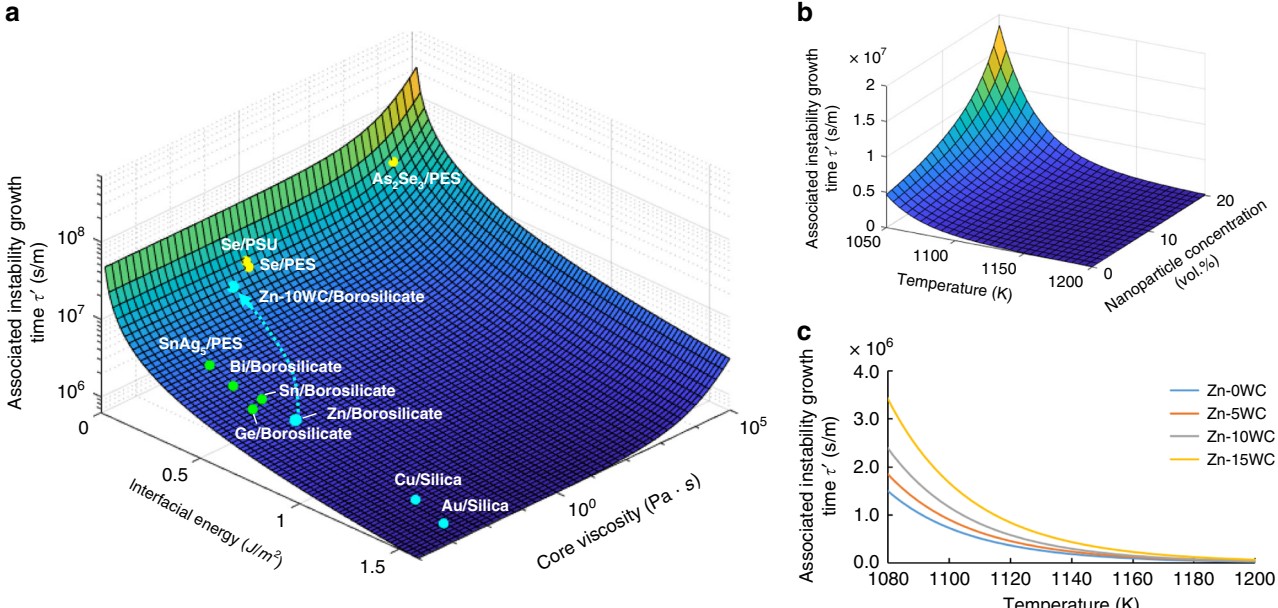

**Fig. 1 Theoretical analysis of fluid instability in thermal drawing in multiple systems. a** Associated instability growth time vs. core material interfacial energy and viscosity under the condition that the cladding viscosity is $10^6$ Pa·s. Data adapted from the literature are specified in Supplementary Table 2. Zinc-WC/borosilicate has been demonstrated to enhance the associated instability growth time by adding nanoparticles to the core material for interfacial energy reduction and viscosity enhancement, as indicated by the cyan arrow. High-melting-temperature metals (Zn, Cu, Au), low-melting-temperature metals (SnAg₅, Sn, Bi, Ge), and semiconductors (Se, As₂Se₃) have been marked on the graph, indicated by cyan, green, and yellow dots, respectively. **b** Theoretical associated instability growth time of Zn-WC nanocomposite as the core material in borosilicate glass cladding, concerning temperature and nanoparticle concentration. **c** The associated instability growth time vs. drawing temperature for Zn-WC/borosilicate system. At a typical drawing temperature of 1080–1100 K, 15 vol.% nanoparticles efficiently enable the fluid instability control.

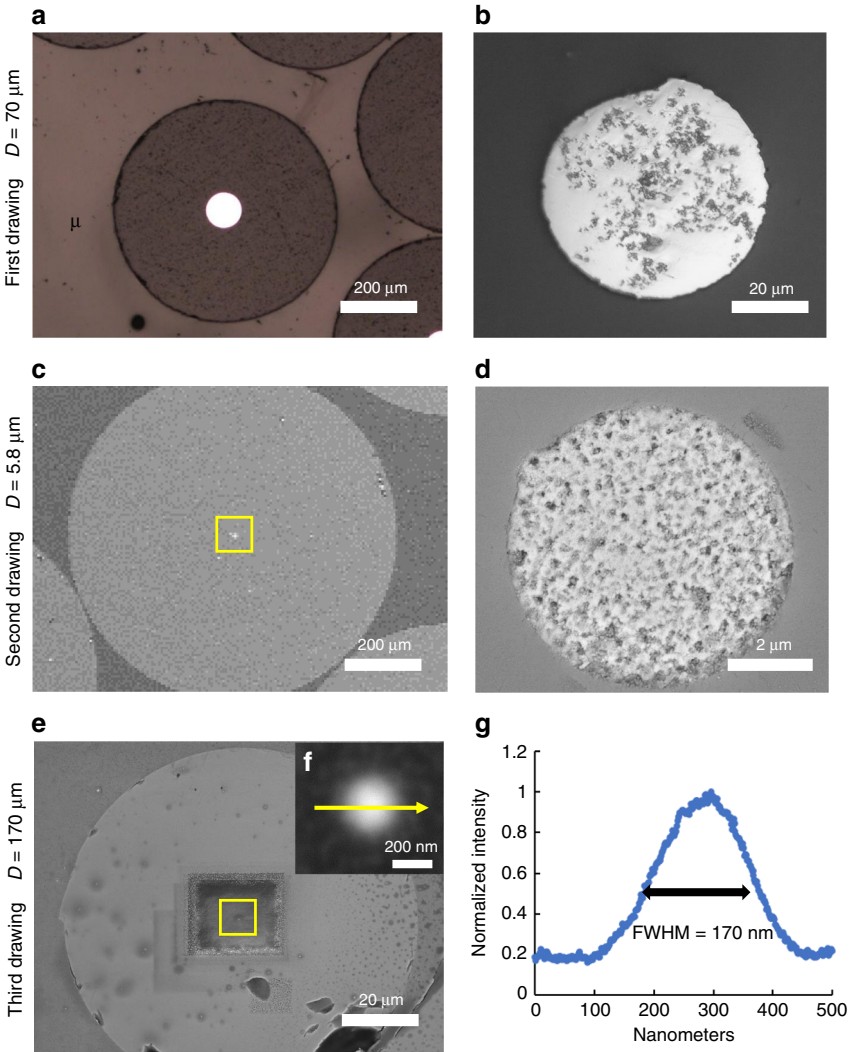

**Fig. 2 Characterization of Zn-WC nanowires embedded in glass cladding. a**, **b** Cross-sectional optical/SEM images of Zn-WC nanocomposite nanowires after the first drawing (70 μm). **c**, **d** Cross-sectional SEM images after the second drawing (5.8 μm). **e**, **f** Cross-sectional SEM images after the third drawing (170 nm). **g** EDS line scanning of Zn-WC nanowire on the cross-section, indicated by the yellow arrow in **f**. The diameter of the nanowire was measured to be 170 nm.

speed and temperature, could successfully produce ultra-long semiconductors nanowires of 10 nm in diameter by thermal drawing, due to their high viscosity and low interfacial energy with cladding. Research has also shown that a high stretching force could result in a certain degree of instability suppression[28].

However, there are still great challenges when the scalable production of metal wires below a few hundreds of nanometers in diameter is needed, because the high interfacial energy between molten metals and glass claddings makes the associated instability growth time too short when the core diameter reaches nanometer scale. An alternative way to realize high-temperature metal nanowires is bottom–up methods, such as chemical synthesis, which is costly, non-scalable, and challenging to apply for industrial manufacturing.

Here we show that the limitations imposed by fluid instabilities with thermally drawn nanowires can be alleviated by adding tungsten carbide (WC) nanoparticles to the metal core to arrive at wire lengths >30 cm with diameters as low as 170 nm. In this study, thermally stable WC nanoparticles were incorporated in zinc to manufacture Zn-WC nanocomposite as the core material in the thermal drawing process. The addition of nanoparticles

promoted the fabrication capability of the nanowire by increasing the viscosity and reducing the interfacial energy between bor-osilicate and Zn[29]. This new mechanism of suppressing fluid instability by nanoparticles not only enables a scalable production of ultra-long metal nanowires but also serves for widespread applications in other fluid-related fields.

## Results

**Nanoparticle-enabled control of fluid instability in thermal drawing of metal wires.** Tomotika dispersion relation is defined as:

$$\tau' = \frac{\tau}{D_{\text{core}}} = \frac{\mu_{\text{clad}}}{\gamma \cdot G\left(x, \frac{\mu_{\text{core}}}{\mu_{\text{clad}}}\right)}, \quad (1)$$

$$x = \frac{\pi D_{\text{core}}}{\lambda}, \quad (2)$$

$$G = \max\left[(1 - x^2) \cdot \Phi\left(x, \frac{\mu_{\text{core}}}{\mu_{\text{clad}}}\right)\right], \quad (3)$$

where $\tau'$ is the associated instability growth time, $\tau$ is the

instability growth time (inverse of the instability growth rate), $D_{core}$ is the diameter of the core, $\mu_{clad}$ and $\mu_{core}$ are the viscosities of the core and cladding, respectively, $\gamma$ is interfacial energy between the core and cladding, $\lambda$ is the varicosity wavelength, and $\Phi$ is a function related to $x$ and viscosity ratio $\mu_{core}/\mu_{clad}$, containing modified Bessel functions. $\tau'$ implies the capability of nanowire fabrication, where larger $\tau'$ refers to a higher capability. In general cases, $x$ falls between 0 and 1. In an ideal case without additional boundary constrains, $G$ defines the maximum instability and the varicosity wavelength at a predefined viscosity ratio. Generally, increasing $\mu_{core}$ results in reduced $G$, thus increasing $\mu_{core}$ (Supplementary Note 1 and Supplementary Fig. 1).

Figure 1a demonstrates the theoretical calculation of the associated instability growth time concerning the core material viscosity and interfacial energy, according to the Tomotika model, through Matlab. Cladding viscosity for all systems is assumed to be $10^6$ Pa·s, a typical value in thermal drawing. Semiconductors with even lower interfacial energy and higher viscosity, like selenium (Se) and $As_2Se_3$, can be fabricated to indefinitely long nanowires with diameters of <100 nm, indicated by yellow dots. Low-melting-temperature metals like tin (Sn), bismuth (Bi), lead (Pb), and indium (In), within polymer claddings, have relatively lower interfacial energy, leading to the manufacturing of indefinitely long wires of 4 μm, indicated by green dots[11]. Metals with high melting temperatures, like zinc (Zn), copper (Cu), silver (Ag), gold (Au), and platinum (Pt), obtain high interfacial energy in glass claddings, indicated by cyan dots. Thus low associated instability growth time results in little success in the fabrication of ultra-long nanowire. It argues for the viability of nanowire fabrication using thermal fiber drawing for materials with high associated instability growth time.

Zn-10 vol.%WC (Zn-10WC)/borosilicate was used as the model material system to demonstrate the fluid instability control by nanoparticles in thermal fiber drawing. The viscosity of Zn-10WC was measured 118 mPa·s at the melting temperature, compared with pure Zn (6 mPa·s), characterized by the modified capillary method, shown in Supplementary Table 1. The contact angle of Zn-10WC/borosilicate was measured to be about 57°, and the corresponding interfacial energy was thus estimated to be 130.6 mJ/m$^2$, shown in Supplementary Note 2 and Supplementary Fig. 2. A typical viable method to enable fluid stabilization of varicose perturbation is by tuning the drawing temperature, but lowering the drawing temperature can enhance the viscosity of cladding material and prohibits high volume production. By applying Zn-WC nanocomposite as the core in thermal fiber drawing, $\tau$ could be significantly increased, due to the increased core viscosity and decreased interfacial energy, shown in Fig. 1a, b. Both core and cladding viscosities are temperature-dependent in this simulation[30,31]. It stabilizes the liquid perturbation during thermal fiber drawing without compromising drawability, as shown in Fig. 1c. At a typical drawing temperature of 1080 K, 15 vol.% WC nanoparticle could theoretically enhance the associated instability growth time to >20 times.

**Nanowire fabrication by thermal drawing.** Zn-10 vol.% WC microwires/nanowires were obtained by thermal fiber drawing (Supplementary Fig. 3) with optimized experimental settings (Supplementary Table 3) for three times, while the drawn fiber was inserted into another borosilicate capillary glass tube for the next drawing. Drawn fibers were characterized by optical microscopy and scanning electron microscopy (SEM) for thickness measurement and continuity verification. The diameter of nanocomposite microwires/nanowires were measured, decreasing

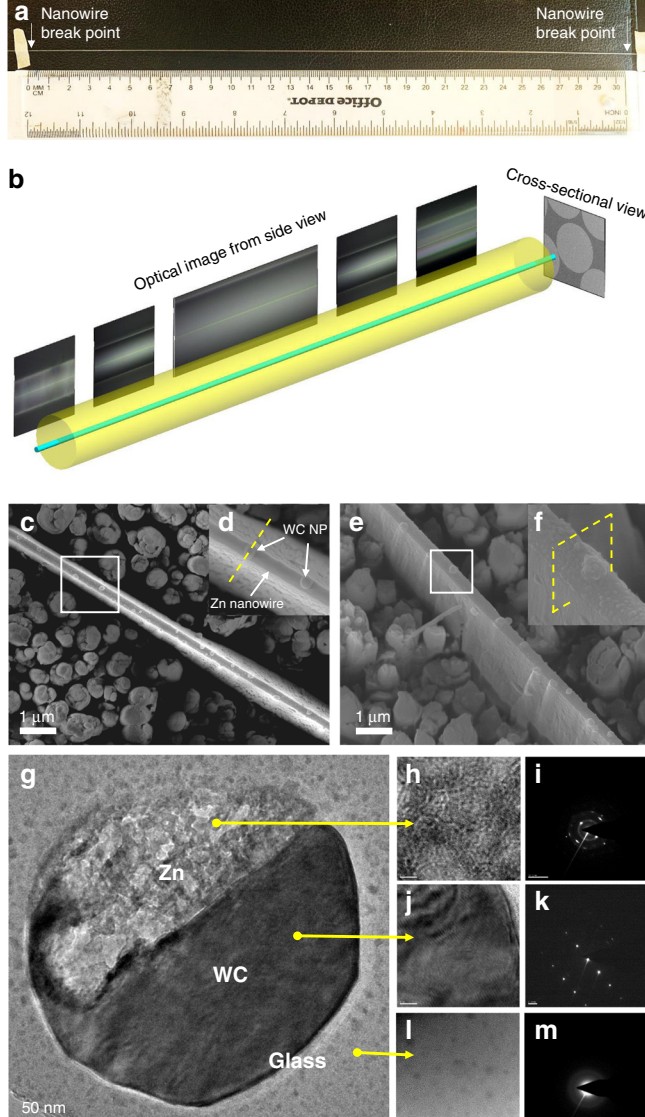

**Fig. 3 Characterization of Zn-WC ultrafine/nanowires integrity. a** 31-cm-long Zn-WC nanowire measured by a ruler. **b** Demonstration of the nanowire continuity characterization by optical microscope. SEM images of the top view (**c**, **d**) and isometric view (**e**, **f**) of Zn-WC nanocomposite nanowire after AOE deep etching. **d**, **f** are the corresponding magnified images. **g** TEM characterization of the cross-sectional nanowire samples, as shown in **d**, **f** by the yellow dashed line. **h**, **j**, **l** are the TEM images of high magnification for Zn and WC nanoparticles and surrounding borosilicate glass, respectively (scale bar 1 nm). **i**, **k**, **m** are the corresponding images of the diffraction pattern, indicating polycrystalline, single crystalline, and amorphous structures.

from 70 μm (Fig. 2a, b) to 5.8 μm (Fig. 2c, d) and eventually to 170 nm (Fig. 2e, f). Zn-WC microwires of 70 μm from the first drawing were polished from the side surface and were characterized by SEM, indicating a homogeneous WC nanoparticle dispersion, shown in Supplementary Fig. 4. Energy-dispersive X-ray spectroscopy (EDS) line scanning was performed to measure the intensity of element Zn across the nanowire cross-section (indicated by the yellow arrow in Fig. 2f) to compensate for the resolution limitation, showing that Zn-WC nanocomposite wire reached a diameter of 171 nm (full width at half maximum) in Fig. 2g.

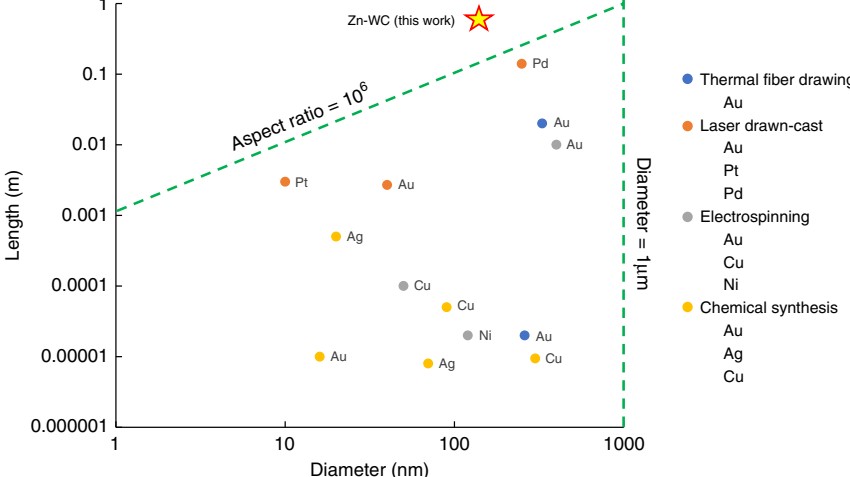

**Fig. 4 Comparison of metal nanowires among different fabrication methods.** The manufacturing methods of high-melting-temperature metal nanowires include thermal fiber drawing, electrospinning, laser drawn-cast process, and chemical synthesis. The green dash lines indicated the limitation that such traditional methods are unable to achieve nano/submicron metal wire fabrication with an aspect ratio of >10⁶. Zn-WC has fundamentally enhanced the instability growth time to realize a much longer nanowire fabrication. Data adapted from the literature are specified in Supplementary Table 4.

**Characterization of nanowire integrity**. Although optical micrographs have resolution limits, it could still be utilized to verify the continuity of the 170 nm Zn-WC nanocomposite wire, since the broken wires, as the results of the Plateau–Rayleigh fluid instability, were usually observed with microscale gaps and satellite metal droplets[32]. The nanowires within the borosilicate cladding were characterized for continuity verification, and the results showed the longest metal nanowire of 31 cm within a glass cladding, as shown in Fig. 3a, b. Even though the curved borosilicate surfaces induced light distortion, the highly bright metal surface could verify the integrity of the Zn nanowires (Supplementary Fig. 5). Furthermore, electrical conductivity was measured for multiple wire segments of 1.5 cm to validate the integrity of the nanowire (Supplementary Fig. 6). Furthermore, anisotropic oxide etching was performed on a small piece of nanowire preserved in glass cladding to expose the bare Zn-WC nanowire. SEM images from the top view and the isometric view were shown in Fig. 3c–f. WC nanoparticles could be observed on the surface of the nanowire, instead of being embedded in the nanowire. The Zn-WC nanocomposite nanowires were also cut using the focused ion beam (FIB) for cross-sectional atomic analysis by transmission electron microscopy (TEM; Fig. 3g). Zn and WC crystalline phase and their corresponding images obtained from fast Fourier transform are shown in Fig. 3h–m.

The successful fabrication of 170 nm nanowire validated the theoretical prediction of the promotion of instability growth time of $\tau$ by using WC nanoparticles to tune viscosity and interfacial energy of Zn melt. $\tau$ of Zn-10WC was calculated to be 1.3 s, which is of the same orders of magnitude as the experimental time, 3.1 s, according to Supplementary Eq. (1). Such theoretical results are not accurate enough without considering the stretching force, but still, sufficiently support the strategy of using nanoparticle to suppress the liquid instability. Besides the inherent property modification of the core metal, it should be noted that additional nanoparticle effects could contribute to the further enhancement of $\tau'$. Nanoparticles at the Zn–borosilicate interface could induce lower interfacial energy locally, and they could act as boundary pinning points to reduce the wavelength of varicosity (Supplementary Note 3 and Supplementary Fig. 7).

As benchmarked in Fig. 4 with metal nanowires manufactured by various methods, including thermal fiber drawing, electrospinning, and laser drawn-cast process, the nanocomposite

nanowire demonstrated in this work exhibits an aspect ratio of $1.8 \times 10^6$. It is noteworthy that the result even exceeds that of metal nanowires produced by chemical synthesis[33–35], which has been one of the most reliable methods for metal nanowire fabrication to date.

In summary, this study presents a nanocomposite approach to fabricate metal nanowires with a high aspect ratio by overcoming fluid instability during thermal drawing. Nanoparticles modified the viscosity of the molten metal and interfacial energy between metal core and cladding when the metal wire diameter is at micrometers. However, once the diameter approaches a few hundreds of nanometers, where the fluid instability becomes almost impossible to control in conventional thermal drawing, nanoparticles can induce low interfacial energy and a pinning effect against the breakdown of the fine molten metal thread inside the glass cladding during thermal drawing. The success in drawing ultra-long metal nanowires by thermal drawing not only offers an exceptional aspect ratio of any metal nanowires produced by any methods so far but also demonstrates the feasibility of fabrication of long metal nanowire by a top-down method.

## Methods

**Fabrication of Zn-WC nanocomposite**. WC nanoparticles (average size of 150 nm, US Research Nanomaterials, Inc.) and Zn powders (50 μm, Fisher Scientific) were first mixed by a mechanical shaker (SK-O330-Pro) at 300 RPM for an hour. The powder mixture was loaded to a cylindrical stainless-steel mold with an inner diameter of 2 cm, followed by cold compaction using a hydraulic press (Carver Laboratory Press, Fred S Carver Inc.) with 58 kN force to obtain Zn-10WC (vol.%) pellets. The Zn-WC nanocomposite pellets were then placed in an alumina crucible and were melted in an electric resistance furnace under argon (Ar) and sulfur hexafluoride (SF₆) gas flow at 450 °C to disperse nanoparticles and eliminate porosity. Molten Zn-WC nanocomposite was then sucked into a 30-cm-long borosilicate glass tube (inner diameter: 1 mm, and outer diameter: 6.5 mm, National Quartz) by a vacuum pump to make the thermal drawing preforms. The 30-cm-long preform was finally fabricated through sealing both ends under vacuum by a high-temperature torch.

**Thermal fiber drawing**. The Zn-WC nanocomposite preform was fed into an electric resistance furnace with a constant feeding speed ($v_f$). The temperature of the furnace was set to 820 °C (1093 K), which is an optimized temperature to obtain the most suitable cladding viscosity (10⁶·⁶ Pa·s) for thermal drawing, after considering the softening point of borosilicate glass and the metal core–furnace temperature difference. The heated preform in the furnace was narrowed down when the temperature increased above the softening point, and the fiber was further pulled down at a constant speed ($v_p$) by a motor-connected wheel. In a

steady-state, the diameter of the drawn-out fiber is determined by the draw-down ratio $D_r$:

$$D_r \equiv \frac{v_f}{v_p} = \left(\frac{d_p}{d_f}\right)^2, \tag{4}$$

where $v_f$, $v_p$, $d_p$, and $d_f$ are the fiber pulling speed, the preform feeding speed, the diameter of a preform, and the diameter of a drawn fiber, respectively. After the first drawing, the metal-in-glass fiber was collected and was inserted into another borosilicate glass tube, working as the preform for the next thermal drawing cycle. During the further drawing, layers of glass cladding could merge, forming a thicker cladding.

**Sample preparation for characterization.** The fiber was first mounted vertically on epoxy and polished to expose the metal core. Then the cross-section was polished, followed by 20-nm-thick Pt deposition and characterized under SEM at low excitation energy, while using FIB to etch off a thin layer on the top to prevent any polishing powder residues from interfering with the characterization. For characterization of bare wires, a combination of hydrofluoric acid (HF) wet etching and AOE oxide etching (STS MESC Multiplex AOE, SPTS Technologies Limited, 90 min) was performed. Due to the chemical reactivity of Zn, nanowires can barely be taken out of the borosilicate cladding massively by any scalable etching method to dissolve the glass claddings. Thus AOE etching was utilized to slowly remove the glass cladding residues after HF wet etching (35 min, 49% aqueous). The bare Zn-WC nanocomposite nanowires were finally exposed for characterization under SEM.

## Data availability

The data that support the findings of this study are available from the corresponding author upon request.

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

## Acknowledgements

We thank Mr. Noah Bodzin for the assistance on FIB cutting and TEM sample preparation at UCLA. This work was supported by National Science Foundation (NSF) (Grant No. CMMI 1449395).

## Author contributions

X.L. conceived the idea and supervised the whole work; I.H. and Z.G. designed the study and conducted the thermal fiber drawing experiment. I.H., Z.G., and C.O.C. did the sample characterization of SEM and TEM; Z.G. conducted Matlab simulation and researched on the nanoparticle pinning theory; Z.G., C.O.C., W.T., and X.L. wrote the manuscript. All authors wrote the paper and discussed the results and commented on the manuscript.

## Competing interests

The authors declare no competing interests.
