## [Peer Review File · Nature Communications]

REVIEWER COMMENTS

Reviewer #1 (Remarks to the Author):

This paper reports interesting observations of being able to draw thin metal fibers out to extreme lengths without incurring instability. However, I find the theoretical explanation of this phenomenon, which makes up the main part of the paper, wholly unconvincing and without much empirical support.

Specifically:

1) I don't see how nanoparticles could have the claimed effect, even qualitatively, and much less so quantitatively. The equation on p. 13 is never used to produce any result that could be tested, for example using a statistical distribution of particles. How are nanoparticles going to restrict instability at nodes? If anything, one could say that particles will 'float' on a long-wave perturbation and enhance instability, if anything.

2) It is claimed that nanoparticles modify bulk parameters such as viscosity or surface tension, but this is never quantified, as would be necessary for a quantitative test.

3) In another paper, Tomotika investigates the role of stretching in inhibiting instability. Perhaps this can explain part of the observed effect?

In conclusion, I advise against publication of this paper in Nature Communications.

Reviewer #2 (Remarks to the Author):

This paper reports an amazing progress on manufacturing of ultralong metal nanowires.

Long metal nanowires have numerous applications. However, making long nanowires is very challenging. Using nanoparticles as fluid stabilizer to overcome the fundamental limit of fluid instability is a very novel solution to this problem.

The mechanisms of the nanoparticle induced stabilization of the thin liquid metal wire were proposed and verified by experimental results.

The record high aspect ratio of 1.8 million achieved in this work is a breakthrough in metal nanowire manufacturing.

Minor comments:

1. Will finer WC nanoparticles lead to even finer nanowire? How does the nanoparticle size influence the aspect ratio of the wire?
2. Was any property of the nanowire tested?
3. Will the volume fraction of nanoparticles influence the drawability of the wire?

Reviewer #3 (Remarks to the Author):

The claim of the paper is a breakthrough in the ability to draw extremely high aspect ratio nanofibers of a metal by means of thermal fiber drawing. By including tungsten carbide (WC) nanoparticles, the authors claim to draw a zinc core inside pyrex glass down to a diameter of approximately 170 nm. This would indeed be both novel and of a high degree of interest to the field.

While it appears they authors have shown at least some length of nanofiber fabricated at a diameter of 170 nm, I am not sure the data shown convinces me that they have fabricated such a nanofiber at a length of 31 cm. Looking at a fiber through a microscope (especially a cylindrical fiber) will result in optical distortions because of the additional lens of the cladding. I have seen in my work that characterizing length continuity this way can often be misleading. A very small diameter core may look continuous, but on further SEM inspection, it is not. The SEM analysis only shows a short length of this nanofiber, so the claim of an aspect ratio of 1.8×10^6 seems unproven to me. There was no reported attempt to measure the conductivity of this nanowire. By measuring the conductivity of a plain pyrex fiber, and then doing the same for a fiber containing the nanowire should lead to differing results, helping to indicate whether the nanowire is continuous.

The authors list as a reason for the continuity being maintained that the WC nanoparticles do not agglomerate. Yet the cross section in Figure 2b shows what appears to be highly agglomerated nanoparticles. It is possible that the agglomeration is ameliorated in subsequent draws, as is somewhat seen in Figure 2d, but it seems highly unlikely that upon even further drawing, all agglomeration disappears. With AOE as a suitable etchant to remove the pyrex cladding, perhaps the authors could try a stack-and-draw approach to produce a fiber with a large number of Zn-WC nanowires. Such a fiber would be easier to characterize in SEM with more wires to observe. This could also help in conductivity measurements to test continuity of the nanowire.

While this paper is very interesting, and could be of great interest to the community, I would recommend further testing and more data before publishing. If the lengths reported cannot be confirmed, then it is possible that this method does not achieve the stated results that would be so beneficial to the community.

RESPONSE TO REFEREES' COMMENTS

Ref. No.: NCOMMS-20-06166

We appreciate the insights from the reviewers and especially are encouraged by the fact that the reviewers saw merit in our work. The comments certainly inspired us to implement careful revisions so as to enhance the quality of our manuscript. Detailed responses to specific comments from the referees are listed as follows.

Response to Referee #1:

Comments:

This paper reports interesting observations of being able to draw thin metal fibers out to extreme lengths without incurring instability. However, I find the theoretical explanation of this phenomenon, which makes up the main part of the paper, wholly unconvincing and without much empirical support.

Specifically:

- 1) I don't see how nanoparticles could have the claimed effect, even qualitatively, and much less so quantitatively. The equation on p. 13 is never used to produce any result that could be tested, for example using a statistical distribution of particles. How are nanoparticles going to restrict instability at nodes? If anything, one could say that particles will 'float' on a long-wave perturbation and enhance instability, if anything.
- 2) It is claimed that nanoparticles modify bulk parameters such as viscosity or surface tension, but this is never quantified, as would be necessary for a quantitative test.
- 3) In another paper, Tomotika investigates the role of stretching in inhibiting instability. Perhaps this can explain part of the observed effect?

Response: Thank you for the comments. We sincerely appreciate your concerns and have revised the manuscript according to your suggestion and recommendations. We carefully reviewed your questions and have improved the manuscript with answers to your doubts below.

(1) From the SEM and TEM images, as shown in Fig. 3c and 3g, some WC particles stayed at the interface between zinc and the glass cladding. Theoretically, due to the minimization of the interfacial energy, WC nanoparticles would actually tend to move to the boundary between the molten Zn and the borosilicate glass [1]. Furthermore, it has been shown that the dispersed nanoparticles can modify the interfacial properties of the ionic liquid [2]. Since the glass cladding is in a highly viscous state, WC nanoparticles at the interface would not be able to move much when compared to the moving liquid zinc. Thus the WC/zinc interfaces would act as the new interface at the points of contacts for the shrinking zinc melt thread during drawing. Since the wettability and interfacial energy between WC particles and Zn melt are favorable (low), the τ will be longer, thus fluid instability will be greatly suppressed (i.e. restricting instability).

$$\tau = \frac{1}{i n} = \frac{D_{core} \cdot \mu_{clad}}{\gamma \cdot G \left(x, \frac{u_{core}}{u_{clad}} \right)}$$

$$x = \frac{\pi D_{core}}{\lambda}$$

$$G = \max \left[\left(1 - x^2 \right) \Phi \left(x, \frac{u_{core}}{u_{clad}} \right) \right]$$

Moreover, the WC nanoparticles at the nanowire surface could act like “permanent nodes” (highly viscous points) for the varicose perturbation waves, namely a pinning effect. Pure liquid threads within a cladding suffer from short instability growth time, which results from a high value of G (besides a high interfacial energy), indicated by the red star in Fig. 5a. Consequently, the nanowires break at the corresponding wavelength, which is directly related to G , shown in Fig. 5b. As nanoparticles could serve as pinning points at the boundary between viscous glass and liquid zinc, the perturbation wavelength was constrained to a significantly small value, resulting in a relatively low G value (indicated by the purple star) and the corresponding long instability growth time (in addition to the contribution from a low interfacial energy), shown in Fig. 5c. Thus, it suggests a pinning effect by WC nanoparticles at the molten Zn – borosilicate interface helps suppress the liquid instability.

Fig.5. Nanoparticle pinning for instability control (a) Plot of function G for the actual viscosity ratio at 950 °C. (b) and (c) are the demonstration varicose perturbation waves with and without nanoparticles. Long-wavelength corresponds to large G value and short associate instability growth time and short-wavelength with nanoparticle pinning as nodes refers to small G value and relatively long associate instability growth time.

(2) Viscosity measurement tests have been performed using a Modified Capillary Method, which is based on the Hagen-Poiseuille formula of capillary flow [3]. The viscosities of pure Zn and Zn-10vol.% WC were measured by

$$\eta = \frac{\pi r^4 t}{8lV} \Delta P$$

through the experiment of constant vacuum pressure suction of the molten metal in a capillary tube. The measured viscosities are shown in Table 1 below.

	Capillary radius (mm)	Capillary length (cm)	Metal volume (mm ³)	Time (s)	Pressure difference (kPa)	Viscosity (mPa·s)
Zn	0.635	30	63.34	0.03	60	6
Zn-10WC	0.635	30	50.67	0.47	60	118

The result validates the higher viscosity induced by dispersed nanoparticles, which helps to overcome fluid instability.

(3) Stretching indeed has been considered a factor in helping stabilize the fluid instability during the thermal drawing process. While a large stretching force during the thermal drawing process could inhibit fluid instability, it limits the reduction ratio consequently. It is extremely difficult, if not impossible, to thermally draw nanoscale metal wires in glass cladding using stretching alone. As a matter of fact, we have experimented with as much as stretching as we could in the system with pure Zn as the core material. But broken wires were observed all the time when the diameter of the zinc melt thread is below a few micrometers.

Reviewer #2 (Remarks to the Author):

This paper reports an amazing progress on manufacturing of ultralong metal nanowires.

Long metal nanowires have numerous applications. However, making long nanowires is very challenging. Using nanoparticles as fluid stabilizer to overcome the fundamental limit of fluid instability is a very novel solution to this problem.

The mechanisms of the nanoparticle induced stabilization of the thin liquid metal wire were proposed and verified by experimental results.

The record high aspect ratio of 1.8 million achieved in this work is a breakthrough in metal nanowire manufacturing.

Minor comments:

1. Will finer WC nanoparticles lead to even finer nanowire? How does the nanoparticle size influence the aspect ratio of the wire?
2. Was any property of the nanowire tested?
3. Will the volume fraction of nanoparticles influence the drawability of the wire?

Response: We sincerely appreciate the favorable comments and have answered the questions point by point, as well as revise the manuscript correspondingly.

1. In this research, we have validated the assumption of nanoparticle-induced fluid instability inhibition. Smaller nanoparticles, higher viscosity in the melt. When using the same volume percentage of nanoparticles, smaller nanoparticles would provide better interfacial area coverage and modification. It would be promising

- to produce even smaller wires. Unfortunately, we could not obtain a smaller WC from the market at this moment. We are certainly interested to try in the future when the smaller WC are available.
2. We have tried to test the mechanical properties of the nanowire under in-situ SEM tensile testing, but unable to obtain results. The zinc nanowires are embedded in the glass, making it more challenging to test. It is quite difficult to etch out the glass while keeping the nanowire intact. We have done electrical conductivity measurement in this revised manuscript.
 3. We have performed only pure Zn and Zn - 10 vol.% WC in the experiment of thermal fiber drawing. Generally speaking, a higher volume fraction (or smaller particle size) would enable the suppression of fluid instability more during the thermal fiber drawing process. However, too many particles could make the solidified zinc nanocomposite wire more brittle, thus potentially breaking off during the solidification stage.

Reviewer #3 (Remarks to the Author):

The claim of the paper is a breakthrough in the ability to draw extremely high aspect ratio nanofibers of a metal by means of thermal fiber drawing. By including tungsten carbide (WC) nanoparticles, the authors claim to draw a zinc core inside pyrex glass down to a diameter of approximately 170 nm. This would indeed be both novel and of a high degree of interest to the field.

While it appears they authors have shown at least some length of nanofiber fabricated at a diameter of 170 nm, I am not sure the data shown convinces me that they have fabricated such a nanofiber at a length of 31 cm. Looking at a fiber through a microscope (especially a cylindrical fiber) will result in optical distortions because of the additional lens of the cladding. I have seen in my work that characterizing length continuity this way can often be misleading. A very small diameter core may look continuous, but on further SEM inspection, it is not. The SEM analysis only shows a short length of this nanofiber, so the claim of an aspect ratio of 1.8×10^6 seems unproven to me. There was no reported attempt to measure the conductivity of this nanowire.

The authors list as a reason for the continuity being maintained that the WC nanoparticles do not agglomerate. Yet the cross section in Figure 2b shows what appears to be highly agglomerated nanoparticles. It is possible that the agglomeration is ameliorated in subsequent draws, as is somewhat seen in Figure 2d, but it seems highly unlikely that upon even further drawing, all agglomeration disappears. With AOE as a suitable etchant to remove the pyrex cladding, perhaps the authors could try a stack-and-draw approach to produce a fiber with a large number of Zn-WC nanowires. Such a fiber would be easier to characterize in SEM with more wires to observe. This could also help in conductivity measurements to test continuity of the nanowire.

While this paper is very interesting, and could be of great interest to the community, I would recommend further testing and more data before publishing. If the lengths reported cannot be confirmed, then it is possible that this method does not achieve the stated results that would be so beneficial to the community.

Response: Thank you for the comments. We sincerely appreciate your comments and have addressed the concerns below.

(1) We would like to acknowledge the fact that the optical microscope does not have the resolution to observe fibers of around 100 nm. However, during our experimental verification of the wire continuity, optical microscopy images indeed showed significant differences in continuous wires and broken wires. First, we are able to verify the continuity of the wires (in microscale) using reverse lighting, as shown in **Error! Reference source not found.** Metal wire embedded fibers were submerged into glycerol to measure the exact wire thickness regardless of the borosilicate glass curve surface as the glycerol and borosilicate shared a similar refractive index. Second, optical microscopy images (top lighting) could distinguish the continuous wire with the observation of the reflective brightness of the metal surface, as shown in **Error! Reference source not found.** Discontinuity could be easily distinguished if the metallic brightness cannot be observed. *As a matter of fact, if there were any nanoscale cracks, they will NOT remain nanoscale as the tensile strain is very large upon solidification of the wires from the drawing high temperature, which will create a microscale gap as we observed all the time in the failed samples.* Also, metal droplets, which were commonly formed due to the Plateau-Reyleigh instability, were not observed in this study [4]. Third, to avoid any nanoscale crack possibility (nanoscale gaps not be observed through optical microscopy), we measured the electrical conductivity of the wire segments to ensure all conductive. Wire segments of 1.5cm length were embedded in epoxy where both ends were exposed with fine grinding and polishing, followed with thick Au-Pd sputtering, and the resistivity of the wires was tested, shown in **Figure 3**. Because of the thick surface coating, resistivity measurement was not accurate, but the linear trend of the voltage-current curve was enough to support the continuity of the wire segments. Multiple segments and repeated tests proved the continuity of the nanoscale wires. Of course, the resistance varies due to the contact quality during the measurements. It should be noted that it is extremely difficult to do the same for the whole 30cm-long wire in one piece due to the fragile, thin glass cladding and difficult handling in the sputtering chamber. But our repeated tests showed the effectiveness of the method.

(2) Regarding the images of Fig.2 in the original manuscript, Fig.2b indicated highly dispersed nanoparticle where agglomeration was not observed in the process. Surface coating of Pd-Au was necessary for SEM imaging of cross-section samples because the glass cladding is non-conductive. However, the coating layer might affect the observation of WC nanoparticle dispersion within the cross-section. Thus, a side view imaging was done to reveal the nanoparticle dispersion within the Zn wires from the first drawing result, shown in **Figure 4**. Fig. 2d indicated the images after FIB cutting of the center section of the wire. Due to the non-conductive property of the glass cladding, Au deposition was required during the imaging of cross-sections and, thus, the Zn surface is not as smooth as polished samples. The darker spots were not nanoparticles, but the non-flat deposition surface. We have substituted using better images, but still, due to cross-

section area reduction, we did not observe nanoparticles from the cross-section, shown in Figure 5.

Figure 1: Optical images of microscale Zn wires, (a) continuous and (b) broken

Figure 2: Optical images of nanoscale Zn wire

Figure 3: (a) embedded fiber segments (tilted) mounted in epoxy, (b) polished surface with wire tip exposed. (c) the resistive measurements between two faces of the epoxy on the sample with nanowire segments.

Figure 4: Zn-WC microwires polished side view image acquired from SEM. WC nanoparticles (bright phases within the microwire) were dispersed homogeneously.

Figure 5: SEM image of the cross section after 2nd drawing. The diameter of the wire was about 2 μm .

References for this response:

1. Xu J, Chen L, Choi H, Konish H, Li X. Assembly of metals and nanoparticles into novel nanocomposite superstructures. *Scientific reports*. 2013;3:1730.
2. Ravera F, Santini E, Loglio G, Ferrari M, Liggieri L. Effect of nanoparticles on the interfacial properties of liquid/liquid and liquid/air surface layers. *The Journal of Physical Chemistry B*. 2006;110(39):19543-51.
3. Gancarz T, Moser Z, Gąsior W, Pstruś J, Henein H. A comparison of surface tension, viscosity, and density of Sn and Sn–Ag alloys using different measurement techniques. *International Journal of Thermophysics*. 2011;32(6):1210-33.
4. Shabahang S, Kaufman J, Deng D, Abouraddy A. Observation of the Plateau-Rayleigh capillary instability in multi-material optical fibers. *Applied Physics Letters*. 2011;99(16):161909.

REVIEWERS' COMMENTS

Reviewer #1 (Remarks to the Author):

The authors have made a significant effort to address my concerns on the theoretical part of the paper, and as a result, the paper is improved significantly. With the additional clarifications given, the proposed mechanism for the suppression of instability appears plausible.

However, there is no real test of the validity of the theory, since the wavelength imposed by the particles is an adjustable parameter.

It would have been nice to have an a-priory estimate of this wavelength, for example by direct observation, or based on the density of nano-particles. However, I acknowledge that this might be technically impossible.

In summary, given the remarkable suppression of instability the authors have achieved, I recommend publication.

Reviewer #2 (Remarks to the Author):

The revised paper is publishable in Nature Communications.

Reviewer #3 (Remarks to the Author):

I believe the authors' responses are sufficient to warrant publication. They have clearly thought about the objections deeply, have some evidence to present as counterpoints, and any remaining possible objections should be raised with the rest of the community. This is only possible through publication.